



# Lidar-based wake tracking for closed-loop wind farm control

Steffen Raach[1], David Schlipf[1], and Po Wen Cheng[1]

[1]Stuttgart Wind Energy (SWE), University of Stuttgart, Allmandring 5B, 70569 Stuttgart, Germany

*Correspondence to:* Steffen Raach (raach@ifb.uni-stuttgart.de)

**Abstract.** This work presents two advancements towards closed-loop wake redirecting of a wind turbine. First, a model-based wake tracking approach is presented which uses a nacelle-based lidar system facing downwind to obtain information about the wake. The method uses a reduced order wake model to track the wake. The tracking is demonstrated with lidar measurement data from an offshore campaign and with simulated lidar data from a SOWFA simulation. Second, a controller for closed-loop wake steering is presented. It uses the wake tracking information to set the yaw actuator of the wind turbine to redirect the wake to a desired position. Altogether, the two approaches enable a closed-loop wake redirection.

## 1 Introduction

In recent years, the focus of control applications in wind energy has shifted more and more to issues in the wind farm. Since wind turbines are growing in size and the available areas for installations are limited, the interactions between wind turbines in a wind farm array are becoming more important. The wind speed in the wake of a wind turbine is reduced with respect to the free stream wind speed. Additionally, the turbulence in the wake is increased. If a wind turbine is hit by a wake from a wind turbine located upwind, the wind turbine produces less power and is faced with higher structural loads because of the increased turbulence, see Borisade et al. (2015). Describing the wake effects and quantifying the decay has been of interest for years. Different models have been developed to address different phenomena, such as the velocity deficit and the increased turbulence intensity. There are empirical models, data driven models, or models which describe the physical behavior in the wake, all varying in complexity and computational effort. Mainly, models with low complexity are steady state models which means they describe the interaction in a static manner and no wake propagation is modeled. Further research is needed to develop control oriented dynamic wake models.

In relation to wind turbine control, the same two goals are valid for wind farm control: 1) maximization of the total power and 2) reduction of the structural loads. These goals were addressed in research with different approaches: 1) axial induction based wind farm control is proposed and investigated and 2) an approach was introduced to redirect the wake. Axial induction control aims at manipulating the axial induction by the blade pitch or torque actuator and steering the wind turbine to a lower production level. This results in a weaker wake deficit and aims at minimizing structural load effects on the downwind wind turbines. The effects on the overall energy capture of the wind farm is not clear yet, see Annoni et al. (2015).

The idea of redirecting the wake by the yaw actuator instead of trying to mitigate its intensity has been discussed in different publications, see Fleming et al. (2014b, a); Gebraad et al. (2014). In simulation studies it was shown that the wake is redirected





up to $0.54$ times the rotor diameter (in seven diameter downwind distance) by yawing the turbine up to $40\,\mathrm{deg}$. Different investigations have shown promising results using this method in open-loop approaches, see Gebraad et al. (2014) and Fleming et al. (2014a).

A major barrier for wind farm control applications is the lack of measurement devices to measure the flow interactions between wind turbines. Further, modeling the three dimensional flow field is not a straight forward approach since the flow is usually described by the Navier-Stokes equations. Lidar can be a useful tool to address the measurement problem in wind farm applications although the limitations of a lidar system always remain and assumptions are necessary to extract the information and exploit the lidar measurement data.

This paper addresses the wind farm control concept of wake redirecting. It aims to enable a closed-loop wake redirecting using lidar measurements to obtain the wake position. The difficulty in wake position definition and measurability is discussed. First, it presents a model-based estimation approach to obtain important quantities for wake redirecting using a nacelle-based lidar system facing downwind. Furthermore, a closed loop controller is In summary, this work presents an entire concept for lidar-based closed-loop wake redirecting.

## 2   Methodology

In order to enable a lidar-based closed-loop wake redirecting within a wind farm, the problem can be divided into two main tasks: 1) the measurement task and 2) the control task. This work focuses mainly on the measurement task but gives also a summary of a solution to the control task, which was presented in Raach et al. (2016). Figure 1 presents the general concept of the closed-loop wake redirecting and the link between measurement task and control task.

### 2.1   Problem formulation for wake-tracking

When talking about wake tracking or a wake center position a main problem exist. There isn't a clear definition of the wake center, moreover, the idea of a wake center comes from time averaging the wake behind a turbine and then characterizing the averaged profile. Having averaged the flow something like a double-Gaussian shape or a Gaussian shape can be observed. From this a wake center can then be defined easily. However, taking a different method of defining the shape, the wake center position could be at a different position although the flow would be the same, see Vollmer et al. (2016). Thus, there isn't a unique wake center definition. This makes a comparison difficult and needs to be considered when comparing results. Furthermore, this means even with full flow field information the wake center is not a measurable quantity and depends on definition.

Considering the task of lidar-based wake tracking then this includes first a reference definition of the wake center and second an estimation method which is used to get the closest estimation of the wake center from the lidar measurement data.

### 2.2   The estimation task

Measuring flow quantities is crucial for enabling a closed-loop controller which want to manipulate the wake quantities. The task of the measurement problem is to provide the necessary quantities for the controller. This means using a measurement




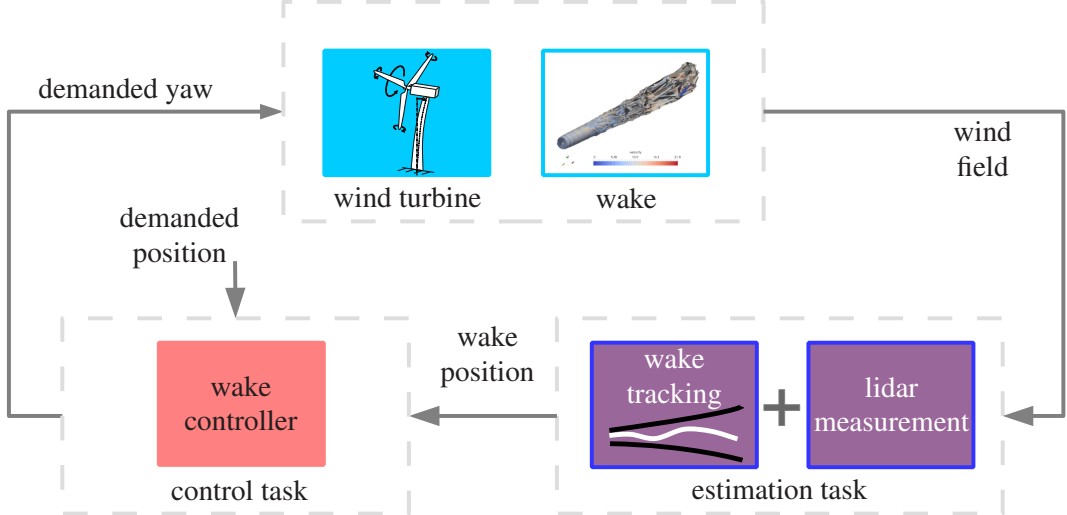

**Figure 1.** The conceptual idea of a closed-loop wake redirecting and its two main tasks: 1) the estimation task addressed in Section 5 and 2) the control task addressed in Section 6.

device, a lidar, and processing the measurement data in such a way that they are useful for the controller. Since the lidar measurement principle has several limitations in providing wind field information an adequate estimation technique is used. This estimation approach is crucial in estimating parameters of the wake and is described in Section 5.

### 2.3 The control task

The second task towards a closed-loop wake redirecting is the control task. Its main challenge is to convert the estimated wake position information and the demanded position to a demanded yaw signal. A feedback controller has to be designed which steers the wake center to the desired position and compensates uncertainties in the models. Since the reaction of a change in the yaw can be measured with a delay, which is due to the wake propagation time, the controller has to be designed in such a way that it can overcome this limitation.

In the following, first, the measurement problem is addressed. A method is presented to estimate wake information from lidar measurement data using a nacelle-based lidar system.

Second, the controller problem is addressed in Section 6. A wake redirecting controller is presented which uses the obtained wake information, namely the wake center position, and steers the wake center using the yaw actuator to a desired position.

The goal of this paper is to present a concept for closed-loop wake redirecting. The in the following described tasks present

a solution to the problem. Therefore, the models can be replaced, modified, or improved but the general concept remains for closed-loop wake redirecting.





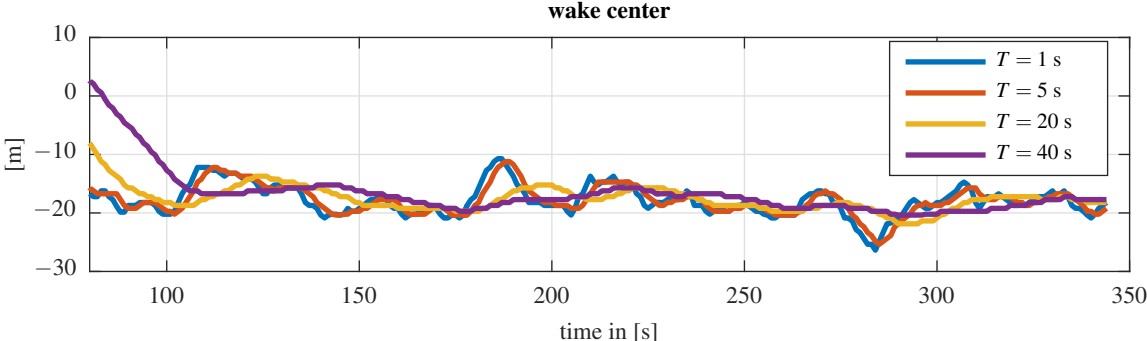

**Figure 2.** Analysis of the impact of different running average filters applied to the flow on the wake center calculation.

## 3 Reference definition and its impact on the estimation task

### 3.1 Wake center definition

As described before, first a reference is needed to be defined. In this work an adaption of the minimum wind power presented in Vollmer et al. (2016) is used. The wake center is defined as the position where the same wind turbine would produce the

least power resulting in

$$\min_{y} \int\limits_{0}^{2\pi} \int\limits_{y}^{R+y} u(r,\phi)^3 r \, \mathrm{d}r\mathrm{d}\phi, \tag{1}$$

where the position of the turbine is described in the polar coordinate system $(r,\phi)$ with the origin at $y$ and $z = 0$ (hub-height). The definition then assumes that the wake center is at $(y,z)$.

The wake center is calculated every time step of the available flow field data which is every second. In addition to Vollmer

et al. (2016) the flow field is time averaged with different time constants. The impact of time averaging is analyzed with different running average filters for the flow and shown in Figure 2. Therefore, a SOWFA simulation with low turbulence level and a mean wind speed of $8 \, \mathrm{m/s}$ is used in which the flow field is sampled and every $1 \, \mathrm{s}$. The wake center clearly converges to a steady value with increasing averaging time $T$.

### 3.2 Problem discussion of lidar-based wake tracking

Compared to other problems in lidar-based wind field reconstruction the problem of wake center estimation is different. Other model based approached in wind field reconstruction like estimation of the rotor-effective wind speed, or estimating $u$ and $v$ wind vector components using lidar measurements like in Schlipf et al. (2012), can first compare to existing quantities. Further, the used models can be used predict line-of-sight velocities of lidar measurements and be directly compared to the real data. Therefore, the model can be used in two directions, estimating and predicting the wind field.



Here, having the wake center defined like in Eq. (1) the prediction of the wind field from a given position is not possible and further a direct comparison of line-of-sight data is not possible. Nevertheless, the wake center position definition seems to be very convenient and is therefore used as reference.

## 4   A simplified wake model for wake tracking

The estimation task addresses the processing and estimation of useful information and provides them to the controller. Since a lidar system has several limitations, the desired quantities, like the wake position, or the wake deficit, are not measurable and have to be estimated from the measurement data. One main limitation of a lidar system is that it only returns the projection of the wind speeds along the direction of the laser beam. This means that a lidar system only provides scalar information of the actual wind vectors. Further, the wind speed is not measured at a certain point but in a volume around the desired measure-

ment location. A solution to this limitations is to implement model-based wind field reconstruction. Wind field reconstruction methods have been developed and used for different applications of lidar system usage in wind energy, for example static two- and three-dimensional, Schlipf et al. (2012), dynamic three dimensional wind field reconstruction methods, see Raach et al. (2014), and approaches for floating lidar systems, Schlipf et al. (2012). Here, the concept of wind field reconstruction is used to obtain information about the wake.

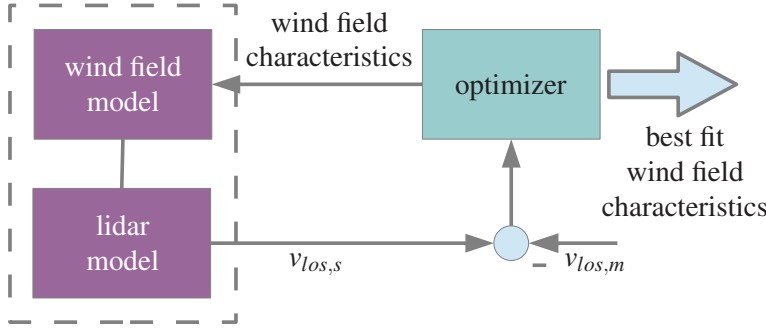

**Figure 3.** The general concept of model-based wind field reconstruction: Estimating the wind field characteristics by fitting simulated lidar measurement data ($v_{los,s}$) to the measured ones ($v_{los,m}$).

The general approach of wind field reconstruction from lidar data is to estimate wind field characteristics from an internal model by fitting simulated lidar measurements to the measured ones. In Figure 3 the basic idea of model-based wind field reconstruction is shown. An optimizer is used to find the best fit for a model of the assumed wind field with the defined lidar configuration. The optimizer minimizes the square error of the modeled (simulated) $v_{los,s}$ and the measured $v_{los,m}$ lidar line-of-sight velocities and returns the estimated wind field parameter.

For the model-based wind field reconstruction an adequate wind field model is crucial. For estimating wake information and tracking the wake, the wind field model has to include a model for the wake in the wind field. Thus, a wake model is necessary





which includes the main wake effects: wake deficit, wake evolution, and wake center displacement. Further, an underlaying wind field model is used. The models are presented in the following section.

## 4.1 Wind field model

Figure 4 shows the subparts of the wind field model: 1) the underlaying wind field, and 2) the wake model.

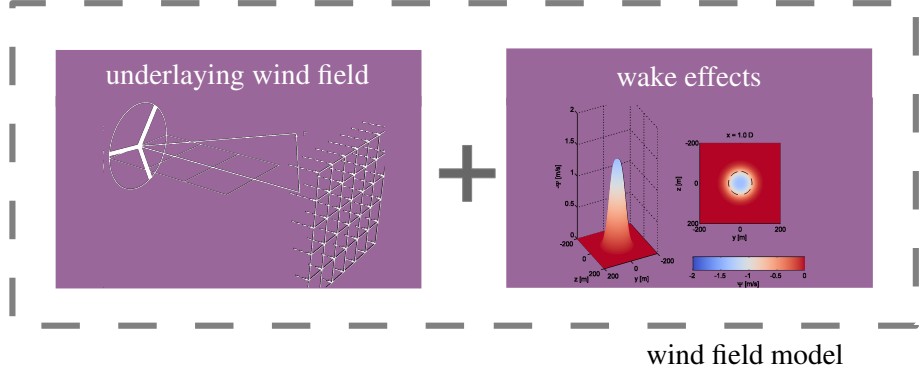

**Figure 4.** The submodels of the wind field model (in the wind coordinate system $W$): 1) the underlaying wind field, and 2) the wake model.

5    The wind field model is described in a wind coordinate system which is denoted by the subscript $W$. It is rotated horizontally with respect to the global inertial coordinate system $I$. The wind speed vector in the $W$-system is transformed in the $I$-system by

$$
\begin{bmatrix} u \\ v \\ w \end{bmatrix}_I = \begin{bmatrix} \cos\alpha & -\sin\alpha & 0 \\ \sin\alpha & \cos\alpha & 0 \\ 0 & 0 & 1 \end{bmatrix} \begin{bmatrix} u \\ v \\ w \end{bmatrix}_W , \tag{2}
$$

where $\alpha$ is the horizontal rotation of the wind field. The underlaying wind field includes the rotor effective wind speed $v_0$
10  and vertical linear shear $\delta_V$. It is assumed that the wind field has only a $u$ component. Thus in the $W$ coordinate system, the underlaying wind field is

$$
\begin{bmatrix} u \\ v \\ w \end{bmatrix}_{i,W} = \begin{bmatrix} v_0 + z_i\delta_V \\ 0 \\ 0 \end{bmatrix} , \tag{3}
$$

where $z_i$ is the height above the ground. This is illustrated in Figure 4 on the left. Further, the wind field is linearly overlaid with the wake model $\Psi$ for the $u$ and $v$ component. Thus, this yields

$$
\begin{bmatrix} u \\ v \\ w \end{bmatrix}_{i,W} = \begin{bmatrix} v_0 + z_i\delta_V + \Psi_{u,i} \\ \Psi_{v,i} \\ 0 \end{bmatrix} . \tag{4}
$$





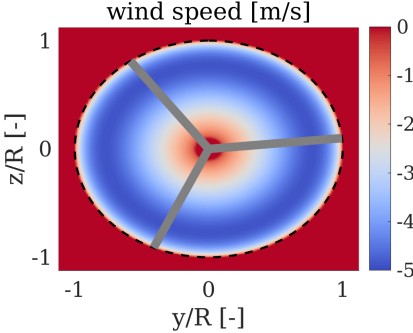

**Figure 5.** The initial wake deficit over the normalized rotor disk.

In the following section, the considered wake effects are described and the wake model is presented.

### 4.1.1 Wake deficit and wake evolution model.

The rotor extracts energy from the wind and converts it into electrical energy. Therefore, the wind speed is reduced behind a wind turbine. Through mixing and energy flow from the surrounding the deficit is cleared over distance. The wake deficit is

modeled with an initial wake deficit at the rotor disk with tip and root losses depending on the energy extraction. In order to get the initial deficit, the energy extraction is mapped by applying Prandtl's root and tip loss function $\Gamma_{\text{Prandtl}}$. Applying the energy conservation assumption yields

$$(v_0 + s\Gamma_{\text{Prandtl}})^2 - (1 - c_P)v_0 = 0, \tag{5}$$

with the power coefficient $c_p$. Solving this equation for $s$ gives the initial wake deficit

$$\Psi_{\text{init}} = s_{\text{solution}}\Gamma_{\text{Prandtl}}. \tag{6}$$

An exemplary initial wake deficit $\Psi_{\text{init}}$ is shown in Figure 5.

The wake is evolving as it moves away from the wind turbine. New energy is flowing from the side and above and the flow is mixed. Physically these dynamics are described via the Navier-Stokes equations. These are partial differential equations and it would be a very complex task to estimate the wake using these equations. However, here an empirical model is used which

models the impulse dissipation. In contrast to other wake models, however, the wake evolution is modeled by a Gaussian shape 2D filter. The 2D filter $\Xi$ depends on the distance $d$ behind the wind turbine

$$\Xi(d, y_i, z_i) = \exp\left(\frac{y_i^2 + z_i^2}{2\sigma_f^2(d)}\right) \tag{7}$$

with

$$\sigma_f(d) = \frac{d\epsilon}{2\sqrt{2\log(2)}} \tag{8}$$



and $y_i$ and $z_i$ the grid points in distance $d$. With the parameter $\epsilon$ the dissipation rate can be set.

Thus, for every distance behind the rotor, the wake can be evaluated using the initial wake deficit $\Psi_{\text{init}}$ and the filter (7). The wake deficit results from the convolution of the initial wake deficit $\Psi_{\text{init}}$ with the filter $\Xi(d, y_i, z_i)$ to

$$\Psi(d, y_i, z_i) = \Xi(d, y_i, z_i) * \Psi_{\text{init}} \tag{9}$$

### 4.1.2 Wake deflection model.

The wake deflection caused by a yaw misalignment $\gamma$ is additionally modeled. The relationship is derived in the study of Jiménez et al. (2010) and was successfully used in an optimization of the yaw angles for a wind farm in Gebraad et al. (2014). The angle of the wake with respect to the main wind direction is

$$\xi(d, c_T, \gamma) = \frac{\xi_{\text{init}}(c_T, \gamma)}{\left(1 + \beta \frac{d}{D}\right)^2}, \tag{10}$$

with the initial angle of the wake at the rotor

$$\xi_{\text{init}}(c_T, \gamma) = \frac{1}{2} \cos^2(\gamma) \sin(\gamma) c_T \tag{11}$$

and the model parameter $\beta$, which defines the sensitivity of the wake deflection to yaw and is here assumed to be known in advance. Further, $c_T$ is the thrust coefficient and $D$ the rotor diameter. Thus, the yaw induced deflection at the downwind position $d$ is according to Gebraad et al. (2014)

$$\delta_{\text{yaw}}(d, c_T, \gamma) = -\xi_{\text{init}}(c_T, \gamma) \frac{D}{30\beta} \left[ 15 \left( 1 - \frac{1}{1 + \frac{2\beta d}{D}} \right) + \xi_{\text{init}}(c_T, \gamma)^2 \left( 1 - \frac{1}{\left(1 + \frac{2\beta d}{D}\right)^5} \right) \right]. \tag{12}$$

The rotation is applied to the wake deficit and yields a $u$ and $v$ component of the wake model,

$$\begin{bmatrix} \Psi_{u,i} \\ \Psi_{v,i} \\ 0 \end{bmatrix}_W = \begin{bmatrix} \cos\xi(d, c_T, \gamma) & -\sin\xi(d, c_T, \gamma) & 0 \\ \sin\xi(d, c_T, \gamma) & \cos\xi(d, c_T, \gamma) & 0 \\ 0 & 0 & 1 \end{bmatrix} \begin{bmatrix} \Psi_i \\ 0 \\ 0 \end{bmatrix}_W. \tag{13}$$

In Figure 6 two different wake situations are shown, the first is a non yawed case and in the second case the turbine is yawed with $\gamma = 25 \deg$. In both cases the underlaying wind field has a constant wind speed of $v_0 = 16\,\text{m/s}$ and no vertical shear.

## 5   The estimation task - model-based wake tracking

As summarized before the estimation task performs the wake tracking using the presented wake model. To perform a lidar-based waked tracking a lidar model is needed. First, the lidar model is presented and then the wake tracking approach is described. Finally, estimation results of two different cases are presented and discussed.




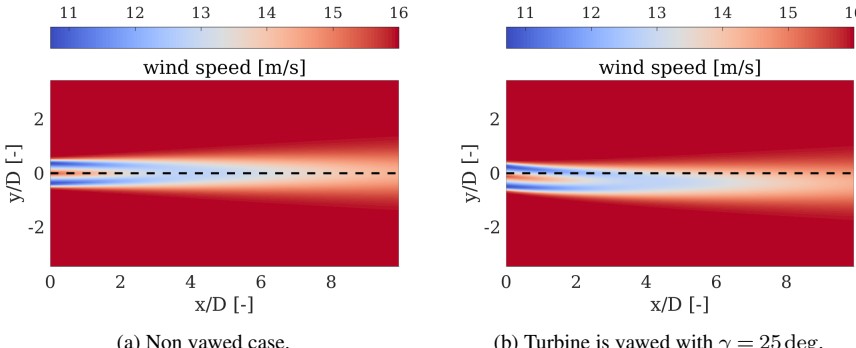

(a) Non yawed case.  (b) Turbine is yawed with $\gamma = 25\,\mathrm{deg}$.

**Figure 6.** Visualization of two wake situations within a constant wind field of $v_0 = 16\,\mathrm{m/s}$, axial induction $a = 0.15$ and dissipation rate $\epsilon = 0.1$.

## 5.1 Lidar model

The lidar measurements can be modeled by a point measurement in the wind field. In the inertial coordinate system this is done by a projection of the wind vector $\begin{bmatrix} u_i & v_i & w_i \end{bmatrix}_I^T$ onto the normalized laser vector in the $i$-th point $\begin{bmatrix} x_i & y_i & z_i \end{bmatrix}_I^T$ with focus distance $f_i = \sqrt{x_{i,I}^2 + y_{i,I}^2 + z_{i,I}^2}$ by

$$v_{los,i} = \frac{x_{i,I}}{f_i} u_{i,I} + \frac{y_{i,I}}{f_i} v_{i,I} + \frac{z_{i,I}}{f_i} w_{i,I}. \tag{14}$$

## 5.2 Model-based wake tracking

As depicted in Figure 3 the model based wind field reconstruction method estimates the model parameter by minimizing the error between measured line-of-sight wind speed $v_{los,m}$ and simulated line-of-sight wind speed $v_{los,s}$. A nonlinear optimization problem is formed for $n$ measurement points. This yields

$$\min_p f(x) = \min_p \begin{bmatrix} (v_{los,m,1} - v_{los,s,1})^2 \\ \vdots \\ (v_{los,m,n} - v_{los,s,n})^2 \end{bmatrix}, \tag{15}$$

where in $p$ all free model parameters are included. The free model parameters are listed in Table 1.

Figure 7 shows one estimation step of the wake tracking from a measurement campaign at the alpha ventus offshore wind farm.

## 5.3 Evaluation and discussion

Figure 7 has already shown that the model fits well for the application and can be applied with real measurement data. In the following, SOWFA (Churchfield and Lee (2012)) is considered as simulation tool. Flow snapshots of a simulation of a single wind turbine are stored and merged to a wind field which is then scanned with a lidar simulator. The lidar scans with a 7x7 grid





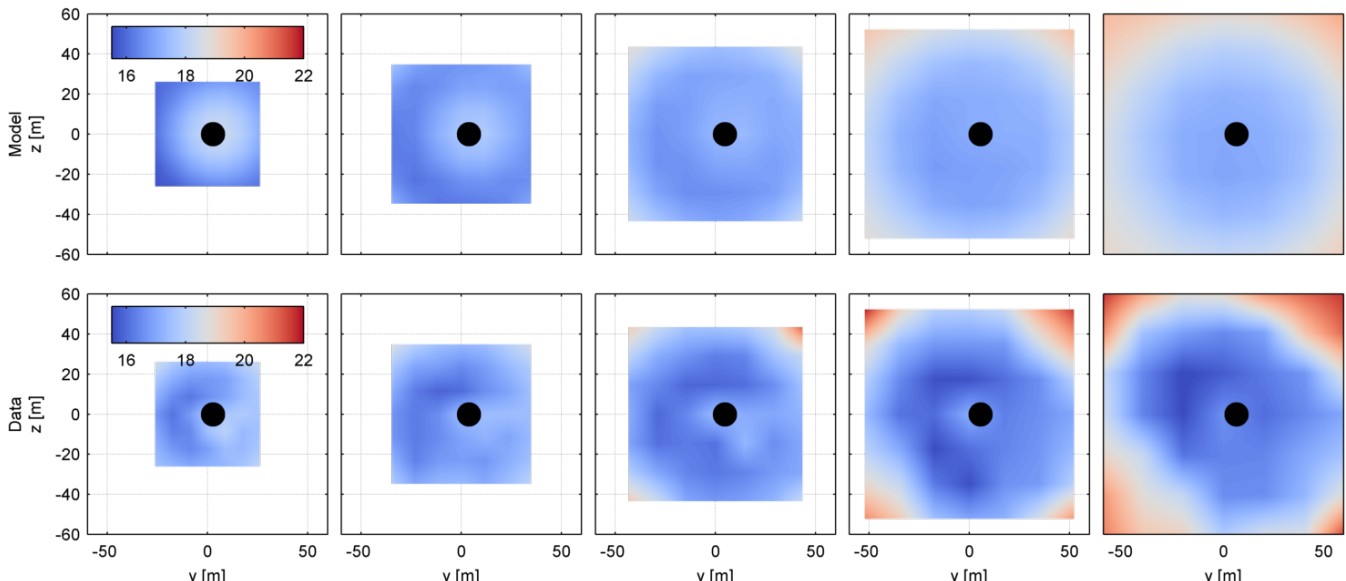

**Figure 7.** A plot of one estimation step of the wake tracking. The simulated lidar measurements in the first row are compared to the measured lidar data in the second row for five distances from $0.6$ to $1.4$ times the rotor diameter (from left to right). The estimated wake center is marked with the black dot.

in five distances from $0.6$ to $1.4$ times the rotor diameter ($D = 126\,\mathrm{m}$). Two different cases are analyzed: First, a case where the turbine is aligned with the wind direction. The estimation results are shown in Figure 8. Second, the turbine is misaligned to deflect the wake. The results of the wake tracking is shown in Figure 9. In both figures the wake center is estimated at the most far scanning distance of $1.4D = 176.4\,\mathrm{m}$. In both cases the method shows the ability of estimating the wake parameter

5 and tracking the wake center.

As mentioned before the wake center positions needs to be calculated using a specific definition and there isn't a direct measurable representation of it. In Figure 10 the lidar-based wake tracking is compared to the wake center estimation using the definition of Eq. (1) without any filtering.

**Table 1.** The free model parameter for the wind field model which are estimated in the optimizer.

| underlaying wind field | | wake model | |
|---|---|---|---|
| $v_0$ | rotor effective wind speed | $c_T$ | thrust coefficient |
| $\delta_V$ | vertical linear shear | $c_P$ | power coefficient |
| | | $\gamma$ | turbine yaw angle |
| | | $\epsilon$ | wake dissipation coefficient |





**Figure 8.** Time results of the wake tracking of a SOWFA simulation. The lidar scanned in a $7x7$ grid in five distances from $0.6D$ to $1.4D$. The wake center is estimated at the most far scanning distance $1.4D = 176.4\,\mathrm{m}$

## 6   The control task

The following closed-loop controller was first presented in Raach et al. (2016) and is recapped here. Then, in a second step having analyzed the wake center displacement from the wake tracking, the filtering is discussed.

As mentioned above, the reaction of the wake to a yaw action can only be measured with a time delay. To control a delayed system, the Smith Predictor approach has been derived and used in many applications. Internal model control is the basic idea of a Smith Predictor.



**Figure 9.** Time results of the wake tracking of a SOWFA simulation. The lidar scanned in a $7x7$ grid in five distances from $0.6D$ to $1.4D$. The wake center is estimated at the most far scanning distance $1.4D = 176.4\,\mathrm{m}$

The presented controller follows the idea of internal model control in which the difference between the actual system output and a predicted output is used within the controller to regulate the system. Therefore, a model is necessary for describing the wake effects in a simplified but sufficient way. It consists of the controller which is a classical proportional-integral controller. Further, an internal model is used which approximates the real system behavior. The wake propagation which exists because the wake flow has to evolve until it reaches the measurement location of the lidar system is approximated with an delay. The time delay $\tau$ varies with respect to the mean wind speed. Finally, a filter is needed to cancel out controller actions which can



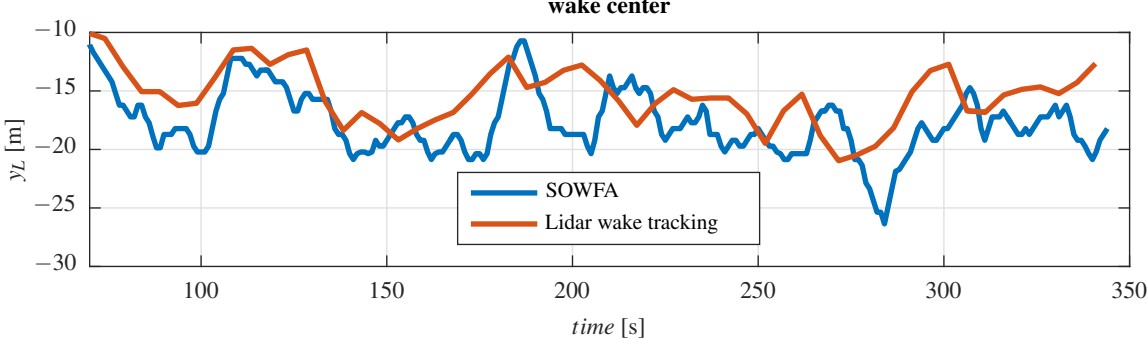

**Figure 10.** Comparison between the wake center estimation (see Eq. (1)) and the lidar-based wake tracking method.

not be observed because of the time difference between control action and measurement location. Figure 11 shows the general concept of the controller.

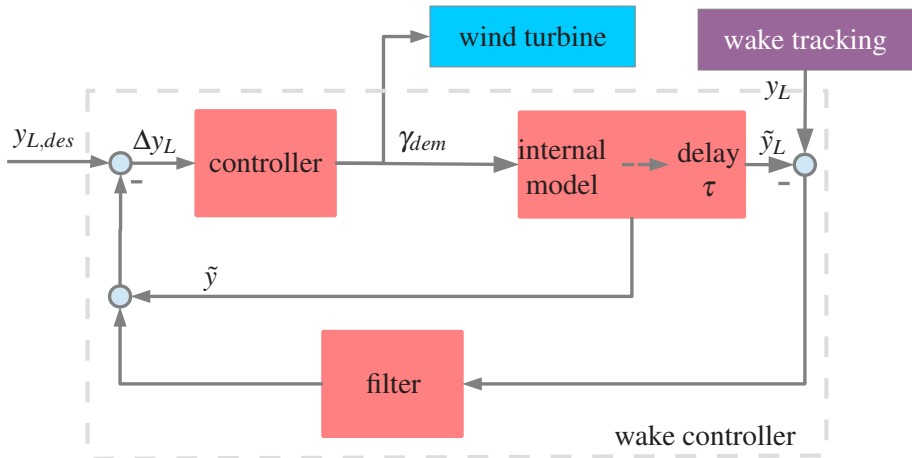

**Figure 11.** The general structure of the wake steering controller: The controller, a simplified wake model and the wake propagation modeled with a delay, and the filter.

## 6.1 Internal wake model of the controller

As depicted in Fig. 11, the wake controller needs an internal model to predict the reaction of the wake to the demanded
5 yaw angle. The internal wake model includes the yaw actuator and the yaw induces wake deflection. For the wake model the assumptions of a constant thrust coefficient, $c_T$, is made.





Altogether, this yields an internal controller model $\widetilde{\Psi}$ of the reality $\Psi$:

$$\widetilde{\Psi} : \begin{cases} \ddot{\gamma} + 2d\omega\dot{\gamma} + \omega^2\gamma = \omega^2\gamma_{dem} & \left| \begin{array}{l} \text{yaw actuator dynamic} \end{array} \right. \\ \tilde{y} = \delta_{\text{yaw}}(d_{\text{Lidar}}, c_{T,\text{const}}, \gamma) & \left| \begin{array}{l} \text{wake deflection model} \end{array} \right. \end{cases} \tag{16}$$

There is a time delay because the wake first needs to evolve to the measurement location:

$$\tilde{y}_L(t) = \tilde{y}(t - \tau). \tag{17}$$

For the controller design, the time delay is approximated using the Pade approximation of time delays.

## 6.2 Controller design

The primal goal of the wake controller is to steer the wake center to a desired point in a defined distance by yawing the wind turbine. As mentioned, this is done using a Smith Predictor. A Smith Predictor uses an internal model to predict the output reaction. Then the predicted wake center position and the filtered error between predicted and measured wake center position is fed back to the controller.

### 6.2.1 Controller

A standard proportional-integral (PI) controller is used. It is designed such that the closed-loop performance with the internal model (16) meets a phase margin of $60\deg$ and a closed-loop bandwidth of $\omega_{\text{CL}} = \frac{1}{2\tau}$. This yields a controller of the form

$$u = K_p\Big(\Delta y_L + \frac{1}{T_i}\int \Delta y_L \mathrm{d}t\Big), \tag{18}$$

with the proportional gain $K_p$ and the time constant $T_i$.

### 6.2.2 Filter

The wake propagation and the caused delay disables a direct measure of a yaw change and because of that one has to filter the measured feedback to prevent non-observable yaw actions. Since the delay $\tau$ is time varying and depends on the mean wind speed the filter has to be adaptable. Therefore, the cutoff frequency of the butterworth low-pass filter is set to $\omega_{\text{filter}} = \frac{\pi}{8\tau}$.

## 6.3 Evaluation and discussion

In the following the wake controller is analyzed. Further, the sensitivity and the complementary sensitivity of the closed-loop system is assessed.

### 6.3.1 Controller analysis

In the following the transfer function of the wake controller is assessed. As shown in Figure 11 the wake controller consists of the internal controller $C$, an internal model $\widetilde{\Psi}$, the time delay approximation $W$ and the filter $F$. Having merged all parts the





wake controller $K$ is

$$K = \frac{F}{(1 + C\,\Psi(1 - F\,W))}. \tag{19}$$

Figure 12 shows the bode analysis of the wake controller $K$. The controller shows integration behavior, starting with $-90\,\mathrm{deg}$ phase.

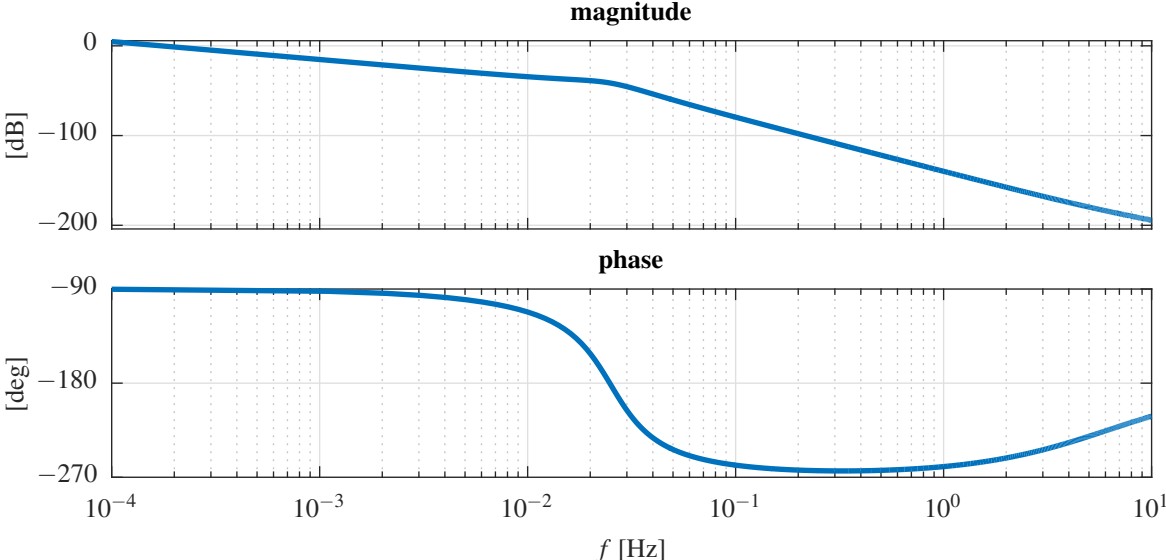

**Figure 12.** Bode analysis of the designed controller $K$.

### 5   6.3.2   Closed-loop analysis

To perform closed-loop analysis the internal controller model $\widetilde{\Psi}$ is transformed to Laplacian space yielding the plant $G$. Then, the sensitivity $S$ and the complementary sensitivity $T$ that are

$$S = \frac{1}{1 + GK} \tag{20}$$

$$T = \frac{GK}{1 + GK}, \tag{21}$$

10   with the controller $K$ are assessed and shown in Figure 13.

## 7   Conclusions

This paper introduces first a method which uses lidar measurements to estimate wind field parameters and enable a tracking of the wake center position. Second, a controller is presented which uses this information to redirect the wake to a desired position.



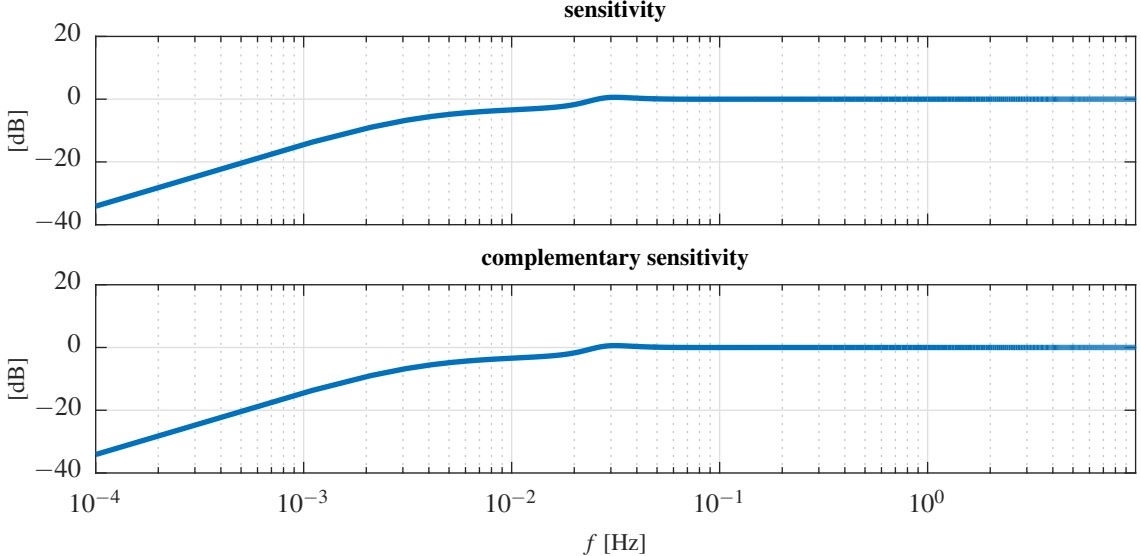

**Figure 13.** Sensitivity $S$ and complementary sensitivity $T$ analysis of the closed-loop system.

In two different cases using simulated lidar measurements of SOWFA simulations, the wake tracking shows promising results in estimating the wake center. The difficulty in wake center position definition is elaborated. A definition is used and the wake tracking results are compared to it.

The challenges of a lidar-based wake redirecting control problem are discussed and an appropriate controller is designed to
5  meet the desired requirements.

This enables the next step towards a closed-loop wake redirecting in an high fidelity simulation tool which is aimed as a next step.

As an outlook, the presented framework of lidar-based closed-loop wake steering offers new possibilities for wind farm control. In a next step, it will be implemented and tested in a high fidelity simulation tool and tested in real time. For the
10  control problem, different controller approaches will be investigated such as $\mathcal{H}_\infty$ controllers or robust controllers. Dynamic estimation techniques as well as other wake estimation models will be used for comparing the ability of tracking the wake and finding the most suitable approach for this task.




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
