# Peer review of "Lidar-based wake tracking for closed-loop wind farm control"

_Wind Energy Science, 2017_

## Referee Comment (RC1) · Anonymous Referee #1 · 1 Feb 2017

**Title :** Lidar-based wake tracking for closed-loop wind farm control

**Authors :** Steffen Raach, David Schlipf, and Po Wen Cheng

**Summary of article :** The authors describe a method of identifying the wake center and a model for wind field reconstruction which is applied both on simulated and measured data to track the near wake and use it as input to a closed-loop yaw control system. The measurements are obtained with a nacelle-mounted lidar. The simulations are performed with SOWFA. A previously proposed controller is revised and proposed to be used alongside the wake tracking algorithm.

**Summary review :** The work seems to be innovative and of value to the scientific community, but the manuscript is extremely hard to follow because of the wording / order of statements. The writing needs to be overhauled so that it can be more accessible. Below some specific suggestions are made to improve clarity.

| Page | Lines | Comment |
|---|---|---|
| 1 | 8 | issues in the wind farm as opposed to...? the individual turbine? |
| | 15 | or > and |
| | 19 | In relation to wind turbine control, the same two goals are valid for wind farm control > The same two goals are valid for both wind turbine and wind farm control |
| | 20 | These goals were addressed in research with different approaches unclear whether you are talking about previous research or your own, please reword it |
| 2 | 1 | in > at |
| | 4-5 | the barrier is not necessarily a lack of devices, but their cost, logistics, etc? |
| | 6-8 | weird sentence, I suggest rewording similar to Lidar can be a useful tool to address the measurement problem in wind farm applications, while bearing in mind the instrument limitations and the assumptions required to extract the information and exploit the lidar measurement data. |
| | 9 | It aims to enable closed-loop wake redirection |
| | 12 | incomplete sentence? |
| Introduction | | Since you are going into so much detail and level of simplicity when you put things in context in the introduction (e.g. flow is modeled by Navier Stokes), you should also briefly explain difference between open vs closed loop control, and why focus on one vs the other. Also, introduction makes it sound like no one else has looked into this before (closed loop wake redirection), is this the case? If so that's fine otherwise you should refer to their work. |
| Figure 1 | | remove a |
| | 20 | exist > exists there isn't a > there is no |
| | 21 | which is a concept based on time averaged profiles of the wake behind a turbine. |
| | 22 | Give a time scale for these averages Having averaged the flow something like a > the language is too informal, reword something like "Averaging the flow yields a (double) Gaussian |

| | | |
|---|---|---|
| | | function for the velocity deficit profile in the horizontal and vertical directions..." |
| | 23-24 | taking a different method of defining the shape, the wake center position could be at a different position although the flow would be the same > also needs rewording, suggestion something like "when different methods are used to define the shape, wake center estimates may be vary under the same flow conditions" |
| | 24-25 | Thus, there isn't a unique wake center definition. This makes a comparison difficult and needs to be considered when comparing results. Suggestion... The absence of a unique wake center definition must be considered when comparing results as it precludes direct comparisons (across different studies?). |
| | 27-28 | Considering the task of lidar-based wake tracking then this includes first a reference definition of the wake center and second an estimation method which is used to get the closest estimation of the wake center from the lidar measurement data. |
| | 30 | which want to > to |
| 3 | 2-3 | "estimate" verb repetition... |
| | 5 | you repeat over and over again "a redirecting" which sounds really off -- either remove "a" or change to a better noun, e.g. redirection |
| | 7 | compensates > compensates for |
| | 8 | can be > is , which is due > due |
| | 10 | first, the measurement problem is addressed. the measurement problem is addressed first. |
| | 11-12 | Keep in same paragraph! |
| | 14 | The in the following described tasks ?? |
| 4 | 3 | As described before, first a reference is needed to be defined. As previously mentioned, it is first necessary to define a reference. (still sounds like an incomplete sentence, a reference what?) |
| | 3-4 | Can you be a bit less concise here? Unclear what the Vollmer work is. Example: The minimum wind power method proposed by Vollmer et al. (2016) is adopted here to identify the wake shape/center....... |
| | 4 | Is all the work 2D? Not yet very clear until this point |
| | 9 | which is every second > sampled at 1 Hz frequency? |
| | 9-10 | In addition to Vollmer et al. (2016) You mean in addition to using the method proposed by Vollmer? |
| | 10 | with different time constants you mean over different running window lengths? over different time intervals? averging time T? |
| | 11-12 | Therefore, a SOWFA simulation with low turbulence level and a mean wind speed of 8 m/s is used in which the flow field is sampled and every 1 s. Therefore? This is a conclusion from something? Needs rewording...ex: The results presented are for a low turbulence (TI=??) SOWFA simulation under a mean (free stream, hub height?) wind speed of 8 m/s... |

| | | |
|---|---|---|
| Figure 2 | | Caption is not descriptive, stand-alone and clear enough. Something like... Time evolution of wake center (meters away from hub? what is negative vs positive?) when different periods T (s) are used to average the flow during the wake center calculation.
Why are first 100 s so different? Is this some model spin up, while the wake is still slowly developing? If so, maybe this data should not be part of the analysis, or this should be acknolwedged somewhere? |
| | 16 | approached > approaches |
| | 17 | can first compare to existing quantities. > can first be compared..... |
| | 16-17 | like estimation of the rotor-effective wind speed, or estimating u and v wind vector components using lidar measurements like in Schlipf et al. (2012), this whole thing should be in parenthesis to make sentence more readable (e.g., estimation of the rotor-effective wind speed, or of u and v wind vector components as in Schlipf et al. (2012)) |
| | 18 | be used predict > be used to predict
after line-of-sight velocities can you put (v_los) so that when it shows up in the next figure the reader is already familiarized with your nomenclature / symbology |
| Figure 3 | | The general concept of model-based wind field reconstruction: Estimating the wind field characteristics by fitting simulated lidar measurement data (vlos,s) to the measured ones (vlos,m).
The general concept of model-based wind field reconstruction, in which the wind field characteristics are estimated by fitting simulated lidar measurement data (vlos,s) to measurements (vlos,m). |
| 5 | 16 | simulated lidar measurements
I am not sure you should call it measurements if they are not measurements! |
| | 19 | What's the " wind field parameter"? |
| | 20-- | This whole paragraph, please rewrite, words and concepts are repeated a lot, very unclear. |
| 6 | 9 | horizontal rotation of the wind field
you mean the wind direction?
Also I'm pretty sure you mean underlying whenever you have underlaying |
| | 13 | ...and the subscript *i* represents...? |
| | 14 | component. Thus, this yields > components, yielding |
| Figure 5 | | If the coordinate system follows the wind turbine reference frame, then what do negative wind speed values mean? Also, does it matter at which downstream distance this is? And is there any yaw misalignment here? Unclear |
| 7 | 4 | the deficit is cleared over distance > the momentum deficit recovers? |
| | 8 | what is s? |
| | 15 | what does " impulse dissipation" mean |
| 8 | 1 | You might want to use D instead of d, or maybe x for the downstream distance, because in equations the little d looks like a derivative, as in Eq 8 I first thought it was derivative of the dissipation. Or maybe just put a multiplication sign there, or the d outside of the fraction multiplying everything... |
| | 19 | by constant you mean steady (constant in time) ? |
| | 7 | the model parameter |

| | | |
|---|---|---|
| | | still confused that THE parameter is? |
| | 12-13 | The way you worded this sentence makes the reader think you want to make a point here. If it's just an example (which at least in this section, it is because now the section is over) then say so--for example:
 An example of an estimation step of the wake tracking from a measurement campaign at the alpha ventus offshore wind farm is shown in Figure 7. |
| Figure 7 | | A plot of
 you don't need to say this is a plot!
 five distances > five downstream distances
 is this looking down or upstream? |
| | 15 | has already shown > shows (you haven't discussed Figure 7 at all) |
| | 17 | merged to a wind field
 what does this mean?
 also use a different symbol in your 7x7, maybe $\times$ wherever it appears in manuscript |
| 10 | 1 | in > at |
| | 4 | most far > furthest (wherever it appears in manuscript)
 the wake parameter , what is this again? |
| | 6 | positions > position
 there isn't a > there is no |
| | | How did you come up with 0.1 for your dissipation? |
| Figure 8 | | Time series of model parameters for wake tracking of simulation data?

 missing a period |
| 11 | 8 | sorry what is "the filtering"? can you be more specific, I don't remember anymore at this point |
| 12 | 5 | an > a |
| Figure 9 | | I assume this is a mistake? I don't understand why it's same caption as above but results are different! |
| 13 | | Figure 11 is talked about in text before Figure 10 so these should be swapped? Actually seems like Figure 10 never comes up?! |
| | 6 | the assumptions of a constant thrust coefficient, $c_T$ , is made.
 the assumption of a constant thrust coefficient is made. |
| 14 | 5 | is this so obvious to the community that it doesn't need a reference? |
| | 2 | what is subscript dem? |
| | 8 | using a Smith Predictor. A Smith Predictor uses > using a Smith Predictor, which uses |
| | 21 | the sensitivity and the complementary sensitivity
 As someone not in controls field I don't understand this. It's weird that in some spots you get into such seemingly unecessary descriptions of things (again, saying the flow is modeled with Navier Stokes for example) but then at other points you assume all your readers will know these concepts? If it's not too difficult, add a line explaining what these concepts mean or refer the reader to some reference. There is a lot of very controls-specific stuff throughout your paper which is fine and great since that's your main topic, but your paper will reach a much broader audience if you make it clearer and more readable to people that do wind research but focus on other aspects, and who may be interested in applying what you've done. |

| 15 | 12 | enable > enables |
|----|-----|---|
| 16 | 5-6 | keep in same paragraph |
|    | 6 | an > a |

---

## Referee Comment (RC2) · Anonymous Referee #1 · 2 Feb 2017

Please download the supplemental material, a 5-page pdf which was uploaded along with the review. It appears to me but if you cannot download it please let me know and I will upload it again.

---

## Short Comment (SC1) · 2 Feb 2017

Could you please specify where you would like to see improvements and where you saw any lack in presentation? I would really appreciate it!

————————————————

---

## Referee Comment (RC3) · Anonymous Referee #2 · 15 Feb 2017

**Title:** Lidar-based wake tracking for closed-loop wind farm control

**Authors:** Steffen Raach, David Schlipf, and Po Wen Cheng

**Summary of article:** The article presents a method for tracking a wake center from lidar measurement data to be used in real time in order to provide an input for a closed-loop wind farm controller. Wake modeling is addressed as it is necessary to take the lidar measured wind field data and transform it into useful information for tracking the wake position downstream of a wind turbine.

**Review Summary:** The article provides a novel approach to tracking the wake center behind a wind turbine using lidar measurements, which will be of value to the wind energy community when trying to develop a closed-loop wind farm controller. The concepts discussed in the paper are well organized and overall has good flow. I was hoping to see more discussion of the controller performance at the end, but this paper is more about the wake tracking than the controller. Perhaps controller performance was discussed in Raach et al. (2014), and could be played up more in this paper to address a reader's desire to see controller performance. It would be nice to see in figures 8 and 9 a comparison to the lidar's tracking of the wake center to the actual wake center. However, defining the wake center is not easy and that is acknowledged by the authors. In practice in the field, defining the wake center is nearly impossible to do anyway as full flow field knowledge is virtually impossible.

| Page: | Line: | Comment: |
|---|---|---|
| 1 | 3 | The tracking is demonstrated… > The wake tracking is demonstrated… |
| 1 | 4 | Spell out the acronym "SOWFA" |
| 1 | 9 & 10 | "The wind speed in the wake of a wind turbine…"

This sentence looks to describe a wake, but seems out of place. Perhaps the wake concept can be introduced in the previous sentence "…installations are limited, the interactions between…" > "…installations are limited, the wake interactions between…" Then this sentence makes more sense. |
| 1 | 11 | If a wind turbine is hit… > If a wind turbine is impacted… |
| 1 | 21 | …is proposed and… > …was… |
| 1 | 22 | …torque actuator and steering the wind turbine to… > …torque actuator and operating the wind turbine at… |
| 1 | 23 | This results in a weaker… > This results in less of a… |
| 1 | 26 | …Fleming et al. (2014b, a); > …Fleming et al. (2014a, b); |
| 2 | 1 | …(in seven diameter… > …(at a seven diameter… |
| 2 | 1 | …by yawing the turbine up to 40 deg.

Is there a reference to back this sentence up? |
| 2 | 12 | …a closed loop controller is In summary,…

It seems there is something missing between "is" and "In" |
| 2 | 20 | …a main problem exist. > …there exists a main problem. |
| 2 | 22 | Having averaged the flow… > After having averaged the flow… |
| 2 | 23 & 24 | However, taking a different method of defining the shape, the wake center position could be at a different position although the flow would be the same, see Vollmer et al. (2016). |

| | | |
|---|---|---|
| | | >
However, if a different method of defining the shape is used with the same flow, the wake center position could be found to be at a different position, see Volmer et al. (2016). |
| 2 | 27 | Considering the task of a lidar-based wake tracking then this includes first a reference definition of the wake center and second…
>
The task of lidar-based wake tracking includes first, a reference definition of the wake center and second,… |
| 2 | 30 | …a closed-loop controller which want to manipulate… > …a closed-loop controller which look to manipulate… |
| 3 | 1 | …device, a lidar, and processing… > …device, such as a lidar, and processing… |
| 3 | 10 | In the following,… > In the following sections,… |
| 3 | 10 – 16 | This should all be one paragraph. |
| 3 | 14 | The in the following described tasks present…

It seems something is missing between "The" and "in" |
| 4 | 3 | …first a reference is needed to be defined. In this work an adaptation… > …first a reference of the wake center is needed to be defined. In this work, an adaptation… |
| 4 | 7 | For equation 1, can you specify the variable $y$ in the following paragraph?  I assume it is the spanwise offset. |
| 4 | 9 | The wake center is calculated every time step…

Can you specify how far downstream the wake center is being calculated here and in figure 2? |
| 4 | 12 & 13 | The wake center clearly converges to a steady value with increasing averaging time T.

This sentence implies that an increasing averaging time is better.  So, just always choose an increasing averaging time is the thought process in my head when I read this.  Perhaps it should be stated that there are adverse effects for choosing an increasing averaging time.  I could see that an increased averaging time would be slower to adjust to a changing wind direction, and so this should be considered when choosing an averaging time to use. |
| 4 | 14 | For section 3.2, the discussion here about comparing between lidar measurements and real data is a little confusing.  I think this is being compared in simulation results.  I think that this section should start by stating that these comparisons are being made in simulation to help a reader to understand these comparisons. |
| 4 | 18 | …the used models can be used… > …the models can be used… |
| 5 | 10 | A solution to this limitations… > A solution to these limitations… |
| 5 | 11 | …applications of lidar system usage in wind energy… > …applications of lidar systems in wind energy… |
| 5 | 12 | …reconstruction methods, see Raach et al… > …reconstruction methods, Raach et al…

To be consistent with the other reference notation in this sentence. |
| 6 | 1 | In the discussion of the main wake effects, I was thinking that wake meandering |

| | | |
|---|---|---|
| | | should be included in this list, but perhaps that falls into the category of wake evolution.  Maybe wake meandering should be its own item in the list, but I do not have a strong opinion one way or another. |
| 6 | 8 | In the discussion with equation 2, I am wondering why do you need to rotate the coordinate system?  I am sure there is a reason, and perhaps you can state why. |
| 7 | 12 & 13 | New energy is flowing from the side and above and the flow is mixed.
>
New energy flows in from the freestream and mixes with the wake. |
| 7 | 15 | In contrast to other wake models, however, … > However, in contrast to other wake models, … |
| 8 | 7 | …optimization of the yaw angles for a wind farm… > …optimization of the yaw angles for a simulated wind farm… |
| 8 | 18 | …non yawed… > …non-yawed… |
| 9 | | For the caption for figure 6, change "Non yawed" to "Non-yawed" |
| 9 | 7 | As depicted in Figure 3 … > As depicted in Figure 3, … |
| 10 | | In figure 7, it would be nice if above each figure in the top row there was a title that specified the downstream distance of each measurement:  0.6 D, ? D, ? D, ? D, 1.4 D.  All I know is 0.6 and 1.4, but the inner distances are not specified. |
| 10 | 2 | Second, the turbine is misaligned…

Could you specify how much the turbine is misaligned? |
| 11 | | In figure 8, the title of the subplot "wake misalignment" is confusing.  Do you mean the turbine's yaw error over time? |
| 12 | | In figure 9, I have the same question about the subplot title as in figure 8. |
| 12 | 5 | …approximated with an delay. > …approximated with a delay. |

---

## Author Comment (AC2) · 22 Mar 2017

Dear Reviewer,

since I am new to this kind of journal, please apologize if I have made mistakes in responding to your feedback.

We highly appreciated it. Attached you find the PTP reply, track changes and the revised manuscript.

Thank you very much for your effort,

Best Steffen

Please also note the supplement to this comment:

http://www.wind-energ-sci-discuss.net/wes-2017-3/wes-2017-3-AC2-supplement.zip
* * *

---

## Author Comment (AC3) · 22 Mar 2017

Dear reviewer,

thank you very much for your feedback and comments. We highly appreciate them and tried to adopt and consider all of them.

Please find attached the PTP reply, the manuscript with track changes (having also included the comments and review of reviewer 1), and a revised manuscript.

Thank you, best Steffen

Please also note the supplement to this comment:
http://www.wind-energ-sci-discuss.net/wes-2017-3/wes-2017-3-AC3-supplement.zip

---

## Author Response (AR1)

**RC1 review**

**Title : Lidar-based wake tracking for closed-loop wind farm control**

Dear reviewer,

We really appreciate your comments and have tried to adopt and consider all of them. Please find below a point-to-point reply. Further, in the supplementary material, a latexdiff is given.

Thank you very much for your effort!

Best
Steffen on behalf of the authors.

*Summary review* : *The work seems to be innovative and of value to the scientific community, but the manuscript is extremely hard to follow because of the wording / order of statements. The writing needs to be overhauled so that it can be more accessible. Below some specific suggestions are made to improve clarity.*

We have considered all of your points, see below, and tried to give more structure. We highly appreciate your effort!

| Page | Lines | Comment | Reply |
|------|-------|---------|-------|
| 1 | 8 | issues in the wind farm as opposed to...? the individual turbine? | Rephrased to clarify |
| | 15 | Or > and | Changed |
| | 19 | In relation to wind turbine control, the same two goals are valid for wind farm control > The same two goals are valid for both wind turbine and wind farm control | Thanks for the suggestion. We adopted it. |
| | 20 | These goals were addressed in research with different approaches unclear whether you are talking about previous research or your own, please reword it | We shortened the sentence and rephrased. |
| 2 | 1 | In > at | Thanks |
| | 4-5 | the barrier is not necessarily a lack of devices, but their cost, logistics, etc? | In the past, there wasn't a measurement device available to measure flow at different locations remotely like a lidar can do. But you are right about the other barriers. We have added them. |
| | 6-8 | weird sentence, I suggest rewording similar to Lidar can be a useful tool to address the measurement problem in wind farm | Thanks! |

| | | | |
|---|---|---|---|
| | | applications, while bearing in mind the instrument limitations and the assumptions required to extract the information and exploit the lidar measurement data. | |
| | 9 | It aims to enable closed-loop wake redirection | Thanks. |
| | 12 | Incomplete sentence? | Sorry about that. We have corrected it. |
| Introduction | | Since you are going into so much detail and level of simplicity when you put things in context in the introduction (e.g. flow is modeled by Navier Stokes), you should also briefly explain difference between open vs closed loop control, and why focus on one vs the other. Also, introduction makes it sound like no one else has looked into this before (closed loop wake redirection), is this the case? If so that's fine otherwise you should refer to their work. | Thank you for the advice. Since my background is control engineering I have assumed a lot. I tried to add a paragraph which briefly explains the advantage of closed-loop wake redirection vs. open-loop. We do not know a publication about lidar-based closed-loop wake redirection at the moment. |
| Figure 1 | | Remove a | Thanks! |
| | 20 | Exist > exists
There isn't a > there is no | Thank you. I have changed. |
| | 21 | which is a concept based on time averaged profiles of the wake behind a turbine. | Thanks. |
| | 22 | Give a time scale for these averages Having averaged the flow something like a > the language is too informal, reword something like "Averaging the flow yields a (double) Gaussian function for the velocity deficit profile in the horizontal and vertical directions..." | We tried to be more specific, thank you for your suggestion. |
| | 23-24 | taking a different method of defining the shape, the wake center position could be at a different position although the flow would be the same > also needs rewording, suggestion something like "when different methods are used to define the shape, wake center estimates may be vary under the same flow conditions" | Thank you. |
| | 24-25 | Thus, there isn't a unique wake center definition. This makes a comparison difficult and needs to be considered when comparing results. Suggestion... | Thank you clarifying. We took you suggestion. |

| | | | |
|---|---|---|---|
| | | The absence of a unique wake center definition must be considered when comparing results as it precludes direct comparisons (across different studies?). | |
| | 27-28 | Considering the task of lidar-based wake tracking then this includes first a reference definition of the wake center and second an estimation method which is used to get the closest estimation of the wake center from the lidar measurement data. | We rephrased it. Thanks. |
| | 30 | Which want to > to | Thanks. |
| 3 | 2-3 | Estimate verb repetition… | We shortened the sentence to be more clear. |
| | 5 | you repeat over and over again "a redirecting" which sounds really off -- either remove "a" or change to a better noun, e.g. redirection | We changed to redirection. |
| | 7 | compensates > compensates for | Thanks. |
| | 8 | can be > is
, which is due > due | Thanks. |
| | 9 | first, the measurement problem is addressed.
the measurement problem is addressed first. | Thanks. |
| | 11-12 | Keep in same paragraph! | Sorry for that. |
| | 14 | The in the following described tasks ?? | We removed parts to state more clearly. |
| 4 | 3 | As described before, first a reference is needed to be defined.
As previously mentioned, it is first necessary to define a reference.
(still sounds like an incomplete sentence, a reference what?) | We changed to be more precisely. |
| | 3-4 | Can you be a bit less concise here? Unclear what the Vollmer work is. Example: The minimum wind power method proposed by Vollmer et al. (2016) is adopted here to identify the wake shape/center……. | We rephrased and adopted your suggestion. |
| | 4 | Is all the work 2D? Not yet very clear until this point | It depends on the method. In our point of view the wake center definition is also not 2D, since all directions are present. |
| | 9 | which is every second > sampled at 1 Hz frequency? | Thanks. |
| | 9-10 | In addition to Vollmer et al. (2016) | We rephrased to be clearer. |

| | | You mean in addition to using the method proposed by Vollmer? | |
|---|---|---|---|
| | 10 | with different time constants you mean over different running window lengths? over different time intervals? averging time T? | Same here. |
| | 11-12 | Therefore, a SOWFA simulation with low turbulence level and a mean wind speed of 8 m/s is used in which the flow field is sampled and every 1 s. Therefore? This is a conclusion from something? Needs rewording...ex: The results presented are for a low turbulence (TI=??) SOWFA simulation under a mean (free stream, hub height?) wind speed of 8 m/s... | Thanks for the comment. We reworded according to you suggestion. |
| Figure 2 | | Caption is not descriptive, stand-alone and clear enough. Something like... Time evolution of wake center (meters away from hub? what is negative vs positive?) when different periods T (s) are used to average the flow during the wake center calculation. Why are first 100 s so different? Is this some model spin up, while the wake is still slowly developing? If so, maybe this data should not be part of the analysis, or this should be acknolwedged somewhere? | Thanks for the suggestion. We adopted the caption. Yes, it comes from the wake development. I have changed the figure according to your suggestion. |
| | 16 | approached > approaches | Thanks. |
| | 17 | can first compare to existing quantities. > can first be compared..... | Thanks. |
| | 16-17 | like estimation of the rotor-effective wind speed, or estimating u and v wind vector components using lidar measurements like in Schlipf et al. (2012), this whole thing should be in parenthesis to make sentence more readable (e.g., estimation of the rotor-effective wind speed, or of u and v wind vector components as in Schlipf et al. (2012) | You are right. Thank you, we adopted it. |
| | 18 | be used predict > be used to predict after line-of-sight velocities can you put (v_los) so that when it shows up in the next figure the reader is already familiarized with your nomenclature / symbology | Thanks. |

| | | | |
|---|---|---|---|
| Figure 3 | | The general concept of model-based wind field reconstruction: Estimating the wind field characteristics by fitting simulated lidar measurement data ($v_{los,s}$) to the measured ones ($v_{los,m}$).
The general concept of model-based wind field reconstruction, in which the wind field characteristics are estimated by fitting simulated lidar measurement data ($v_{los,s}$) to measurements ($v_{los,m}$). | Thanks. This makes it more clearer. |
| 5 | 16 | simulated lidar measurements
I am not sure you should call it measurements if they are not measurements! | Thank you. We have changed it here. |
| | 19 | What's the " wind field parameter"? | We specified. We meant the model parameter (e.g. wake center, wake decay, wake deficit, etc.) |
| | 20-- | This whole paragraph, please rewrite, words and concepts are repeated a lot, very unclear. | We have rewritten. |
| 6 | 9 | horizontal rotation of the wind field you mean the wind direction?
Also I'm pretty sure you mean underlying whenever you have underlaying | Yes, we mean aligned with the wind direction.
Yes, sorry about that! |
| | 13 | ...and the subscript i represents...? | We explained. |
| | 14 | component. Thus, this yields > components, yielding | Thanks. |
| Figure 5 | | If the coordinate system follows the wind turbine reference frame, then what
do negative wind speed values mean? Also, does it matter at which downstream distance this is? And is there any yaw misalignment here? Unclear | We specified the conditions. |
| 7 | 4 | the deficit is cleared over distance > the momentum deficit recovers? | Yes, thanks. |
| | 8 | What is s? | s*Gamma gives the solution for the initial wake deficit. There is no meaning for s -> one could see it as local gain. |
| | 15 | what does " impulse dissipation" mean | We meant wake recovery. |
| 8 | 1 | You might want to use D instead of d, or maybe x for the downstream distance, because in equations the little d looks like a derivative, as in Eq 8 I | Thanks. The multiplication sign helped. |

| | | | |
|---|---|---|---|
| | | first thought it was derivative of the dissipation. Or maybe just put a multiplication sign there, or the d outside of the fraction multiplying everything... | |
| | 19 | by constant you mean steady (constant in time) ? | Mean wind speed is meant |
| 9 | 7 | the model parameter still confused that THE parameter is? | Changed |
| | 12-13 | The way you worded this sentence makes the reader think you want to make
a point here. If it's just an example (which at least in this section, it is because now the section is over) then say so--for example:
An example of an estimation step of the wake tracking from a measurement campaign at the alpha ventus offshore wind farm is shown in Figure 7. | Thanks. |
| Figure 7 | | A plot of
you don't need to say this is a plot!
five distances > five downstream distances
is this looking down or upstream? | Thanks ;)
We clarified the setup. |
| | 15 | has already shown > shows (you haven't discussed Figure 7 at all) | Thanks. |
| | 17 | merged to a wind field
what does this mean?
also use a different symbol in your 7x7, maybe $\times$ wherever it appears in manuscript | We removed the unclear part and used the times symbol. |
| 10 | 1 | In >at | Thanks |
| | 4 | most far > furthest (wherever it appears in manuscript)
the wake parameter , what is this again? | Thanks. We have added some lines before. So the parameter question should be clear. |
| | 6 | positions > position
there isn't a > there is no | Thank you. |
| | | How did you come up with 0.1 for your dissipation? | It is the result of the model fit. |
| Figure 8 | | Time series of model parameters for wake tracking of simulation data?
missing a period | We specified the conditions. |
| 11 | 8 | sorry what is "the filtering"? can you be more specific, I don't remember anymore at this point | Removed the sentence, since it isn't necessary here. It is only confusing. |
| 12 | 5 | An > a | Thanks |

| | | | |
|---|---|---|---|
| Figure 9 | | I assume this is a mistake? I don't understand why it's same caption as above but results are different! | Yes! As mentioned before, we have specified the conditions. |
| 13 | | Figure 11 is talked about in text before Figure 10 so these should be swapped? Actually seems like Figure 10 never comes up?! | It was mentioned in the text. Before Sect. 6. |
| | 6 | the assumptions of a constant thrust coefficient, c T , is made. the assumption of a constant thrust coefficient is made | Thanks. |
| 14 | 5 | is this so obvious to the community that it doesn't need a reference? | A reference is given. |
| 15 | 2 | what is subscript dem? | A description is added. |
| | 8 | using a Smith Predictor. A Smith Predictor uses > using a Smith Predictor, which uses | Thanks. |
| | 21 | the sensitivity and the complementary sensitivity As someone not in controls field I don't understand this. It's weird that in some spots you get into such seemingly unecessary descriptions of things (again, saying the flow is modeled with Navier Stokes for example) but then at other points you assume all your readers will know these concepts? If it's not too difficult, add a line explaining what these concepts mean or refer the reader to some reference. There is a lot of very controls-specific stuff throughout your paper which is fine and great since that's your main topic, but your paper will reach a much broader audience if you make it clearer and more readable to people that do wind research but focus on other aspects, and who may be interested in applying what you've done. | We have added a reference. Sorry about that. It is very difficult to address and assume the right audience. Our assumption was to address someone who has basic knowledge in control theory. |
| 15 | 12 | enable > enables | Thanks |
| 16 | 5-6 | Keep in same paragraph | Changed. |
| | 6 | An > a | Thanks! |

**Title : Lidar-based wake tracking for closed-loop wind farm control**

Dear reviewer,

We really appreciate your comments and have tried to adopt and consider all of them. Please find below a point-to-point reply. Further, in the supplementary material, a latexdiff is given. (Having already considered the review of reviewer 1)

Thank you very much for your effort!

Best
Steffen on behalf of the authors.

*Summary review* : *The article provides a novel approach to tracking the wake center behind a wind turbine using lidar measurements, which will be of value to the wind energy community when trying to develop a closed-loop wind farm controller. The concepts discussed in the paper are well organized and overall has good flow. I was hoping to see more discussion of the controller performance at the end, but this paper is more about the wake tracking than the controller. Perhaps controller performance was discussed in Raach et al. (2014), and could be played up more in this paper to address a reader's desire to see controller performance. It would be nice to see in figures 8 and 9 a comparison to the lidar's tracking of the wake center to the actual wake center. However, defining the wake center is not easy and that is acknowledged by the authors. In practice in the field, defining the wake center is nearly impossible to do anyway as full flow field knowledge is virtually impossible.*

Thank you for your review. You are completely right, this paper covers more the estimation task and the 2016 ACC and also my current work focus on the control part. I will mention it in the beginning of the control part and in the conclusion. Figure 10 gives exactly what you asked for. You are completely right, however, when talking about the wake center definition and comparability challenge.

| Page | Lines | Comment | Reply |
|------|-------|---------|-------|
| 1 | 3 | The tracking is demonstrated… > The wake tracking is demonstrated… | Thanks |
| 1 | 4 | Spell out the acronym "SOWFA" | Changed. |
| | 9-10 | "The wind speed in the wake of a wind turbine…"
This sentence looks to describe a wake, but seems out of place. Perhaps the wake concept can be introduced in the previous sentence "…installations are limited, the interactions between…" >
"…installations are limited, the wake interactions between…" Then this sentence makes more sense. | Thanks for the suggestion. I considered it. |

| | 11 | If a wind turbine is hit… > If a wind turbine is impacted… | Thanks. |
|---|---|---|---|
| | 21 | …is proposed and… > …was… | Has been rephrased. |
| | 22 | …torque actuator and steering the wind turbine to… > …torque actuator and operating the wind turbine at… | Thank you. We adopted it. |
| | 23 | This results in a weaker… > This results in less of a… | Thanks. |
| | 26 | …Fleming et al. (2014b, a); > …Fleming et al. (2014a, b); | Thanks. This was a strange behavior of the bibtex package |
| 2 | 1 | …(in seven diameter… > …(at a seven diameter… | Thanks. |
| | 1 | …by yawing the turbine up to 40 deg. Is there a reference to back this sentence up? | Added. |
| | 12 | …a closed loop controller is In summary,… It seems there is something missing between "is" and "In | Our fault. We have corrected it. |
| | 20 | …a main problem exist. > …there exists a main problem. | Thanks. |
| | 22 | Having averaged the flow… > After having averaged the flow… | Rephrased. |
| | 23-24 | However, taking a different method of defining the shape, the wake center position could be at a different position although the flow would be the same, see Vollmer et al. (2016). | Rephrased. |
| | 27 | Considering the task of a lidar-based wake tracking then this includes first a reference definition of the wake center and second… > The task of lidar-based wake tracking includes first, a reference definition of the wake center and second,… | Thanks. We adopted it. |
| | 30 | …a closed-loop controller which want to manipulate… > …a closed-loop controller which look to manipulate… | We rephrased and tried to make it clearer. |
| 3 | 1 | …device, a lidar, and processing… > …device, such as a lidar, and processing… | Thank you. |
| | 10 | In the following,… > In the following sections,… | Thanks. |

| | | | |
|---|---|---|---|
| | 10-16 | This should all be one paragraph | Ok |
| | 14 | The in the following described tasks present…
 It seems something is missing between "The" and "in" | We rephrased it. |
| 4 | 3 | …first a reference is needed to be defined. In this work an adaptation… > …first a reference of the wake center is needed to be defined. In this work, an adaptation… | We removed parts and rephrased the beginning. |
| | 7 | For equation 1, can you specify the variable y in the following paragraph? I assume it is the spanwise offset. | Thank you. We missed that. But we prefer to use "lateral offset". |
| | 9 | The wake center is calculated every time step…
 Can you specify how far downstream the wake center is being calculated here and in figure 2? | Thanks, good point! |
| | 12-13 | The wake center clearly converges to a steady value with increasing averaging time T.
 This sentence implies that an increasing averaging time is better. So, just always choose an increasing averaging time is the thought process in my head when I read this. Perhaps it should be stated that there are adverse effects for choosing an increasing averaging time. I could see that an increased averaging time would be slower to adjust to a changing wind direction, and so this should be considered when choosing an averaging time to use | Very good point, we have added a sentence like you suggested. |
| | 14 | For section 3.2, the discussion here about comparing between lidar measurements and real data is a little confusing. I think this is being compared in simulation results.
 I think that this section should start by stating that these comparisons are being | We have added something at the beginning of section 3. |

| | | | |
|---|---|---|---|
| | | made in simulation to help a reader to understand these comparisons. | |
| | 18 | …the used models can be used… > …the models can be used… | Thanks. |
| 5 | 10 | A solution to this limitations… > A solution to these limitations… | Thanks. |
| | 11 | …applications of lidar system usage in wind energy… > …applications of lidar systems in wind energy… | Thanks. |
| | 12 | …reconstruction methods, see Raach et al… > …reconstruction methods, Raach et al… To be consistent with the other reference notation in this sentence. | Thanks. |
| 6 | 1 | In the discussion of the main wake effects, I was thinking that wake meandering should be included in this list, but perhaps that falls into the category of wake evolution. Maybe wake meandering should be its own item in the list, but I do not have a strong opinion one way or another. | Since it is not modeled in the reduced order model, we haven't mentioned it. Since the model is used for identification the meandering DOF is not necessary at the moment, but could be considered if necessary. |
| | 8 | In the discussion with equation 2, I am wondering why do you need to rotate the coordinate system? I am sure there is a reason, and perhaps you can state why. | It is just a convention to introduce different coordinate systems for wind, lidar, turbine. It gives the freedom to yaw the turbine, or consider a misaligned wind field in the reconstruction. |
| 7 | 12-13 | New energy is flowing from the side and above and the flow is mixed. > New energy flows in from the freestream and mixes with the wake. | Thank you, good point! |
| | 15 | In contrast to other wake models, however, … > However, in contrast to other wake models, … | Thanks. |
| 8 | 7 | …optimization of the yaw angles for a wind farm… > …optimization of the yaw angles for a simulated wind farm… | Ok. |
| 8 | 18 | …non yawed… > …non-yawed… | Thanks. |

| 9 | | For the caption for figure 6, change "Non yawed" to "Non-yawed" | Done. |
|---|---|---|---|
| 9 | 7 | As depicted in Figure 3 … > As depicted in Figure 3, … | Thanks. |
| 10 | | In figure 7, it would be nice if above each figure in the top row there was a title that specified the downstream distance of each measurement: 0.6 D, ? D, ? D, ? D, 1.4 D. All I know is 0.6 and 1.4, but the inner distances are not specified. | We have added the distances in the caption. |
| | 2 | Second, the turbine is misaligned… Could you specify how much the turbine is misaligned | It is done. |
| 11/12 | | In figure 8/9, the title of the subplot "wake misalignment" is confusing. Do you mean the turbine's yaw error over time? | Yes. I will correct. |
| 12 | 5 | …approximated with an delay. > …approximated with a delay. | Thanks. |

**Lidar-based wake tracking for closed-loop wind farm control**

Steffen Raach[1], David Schlipf[1], and Po Wen Cheng[1]

[1]Stuttgart Wind Energy (SWE), University of Stuttgart, Allmandring 5B, 70569 Stuttgart, Germany

*Correspondence to:* Steffen Raach (raach@ifb.uni-stuttgart.de)

**Abstract.** This work presents two advancements towards closed-loop wake redirecting of a wind turbine. First, a model-based wake tracking approach is presented which uses a nacelle-based lidar system facing downwind to obtain information about the wake. The method uses a reduced order wake model to track the wake. The wake tracking is demonstrated with lidar measurement data from an offshore campaign and with simulated lidar data from a  simulation with the Simulator fOr Wind Farm Applications (SOWFA). Second, a controller for closed-loop wake steering is presented. It uses the wake tracking information to set the yaw actuator of the wind turbine to redirect the wake to a desired position. Altogether, the two approaches enable a closed-loop wake redirection.

**1 Introduction**

In recent years,  wind farm control has gained more and more  importance in the wind  energy control community, since wind turbines in a wind farm  can interact by their flow. The wake interaction can result in less power compared to a fee-stream operation and can result in higher structural load of the downstream turbine due to higher turbulence in the flow and possible partial wake impingements. 
[revised manuscript text omitted]

---

## Referee Report (RR1)

**Title :** Lidar-based wake tracking for closed-loop wind farm control
**Authors :** Steffen Raach, David Schlipf, and Po Wen Cheng

striked text: remove
highlighted text: see comment

**Section 1**

1. due to interactions between individual wind turbines in a wind farm
2. of interest to whom?
3. velocity deficit and turbulence intensity are not "phenomena"
4. have
5. a lower
6. contains
7. and
8. ambiguous, makes it sound like it's the cost/availability of wind turbines -- reword
9. A model-based estimation approach is presented and used to obtain...
10. , and a closed loop controller is...

Combine last two paragraphs of introduction.

**Section 2**

11. redirection
12. two main tasks must be considered
13. redirection
14. between the measurement and control tasks
15. The main issue in the context of wake tracking algorithms is that no clear definition exists for the wake center.

Refer the reader to references below for a review of wake center estimation methods.

Doubrawa, Paula, Rebecca J. Barthelmie, Hui Wang, and Matthew J. Churchfield. "A Stochastic Wind Turbine Wake Model Based on New Metrics for Wake Characterization." *Wind Energy* 20, no. 3 (March 1, 2017): 449–63. doi:10.1002/we.2015.
Howland, Michael F., Juliaan Bossuyt, Luis A. Martinez-Tossas, Johan Meyers, and Charles Meneveau. "Wake Structure of Wind Turbines in Yaw under Uniform Inflow Conditions." *arXiv:1603.06632 [physics]*, March 21, 2016. http://arxiv.org/abs/1603.06632.

Combine last two paragraphs of Section 2.1.

16. not sure what you mean by "manipulate" wake quantities?
17. as
18. its desired value
19. to
20. put this in parenthesis

21. while providing

**Section 3**

22. through the use of simulation data, which can cover a larger area at a higher spatial and temporal resolution than measurements would be able to provide.
23. oriented
24. the downstream position where a hypothetical turbine of identical characteristics and yaw angle would produce the least power.
25. y=0
26. window
27. obtained
28. turbulence intensity
29. it is not microsecond^-1 but instead meter second^-1 so put a space between m and s
30. The problem of wake center estimation is different from other problems in lidar-based wind field reconstruction.
31. When the wake center is defined as
32. neither is
33. "seems to be very convenient" is not scientific language

Combine last two paragraphs of Section 3.2.

**Section 4**

34. directly measurable
35. wind lidar systems
36. they return
37. Most of your citations are of people that you've worked with or of your immediate network, it's best to diversify this a bit to show you've done your research and consider the body of work in the field while performing your analyses. Have you seen the reference below for model simulations of lidar measurements in the wake?

Lundquist, J. K., M. J. Churchfield, S. Lee, and A. Clifton. "Quantifying Error of Lidar and Sodar Doppler Beam Swinging Measurements of Wind Turbine Wakes Using Computational Fluid Dynamics." *Atmos. Meas. Tech.* 8, no. 2 (February 23, 2015): 907–20. doi:10.5194/amt-8-907-2015.

38. The "s" of Figure 3 is chopped off
39. example (never exemplary)

**exemplary |igˈzemplərē|**
adjective
**1** serving as a desirable model; representing the best of its kind: *an award*

*for exemplary community service*.

**2** (of a punishment) serving as a warning or deterrent: *exemplary sentencing may discourage the ultraviolent minority*.

• Law (of damages) exceeding the amount needed for simple compensation.

40. evolves
41. it's not really because they are PDOs that it is difficult to use Navier-Stokes for your purposes. It's because of the non-linear terms and the closure problem that arises from the non-linearity, right?
42. the rotor diameter D comes up several times before so please define it at its first ocurrence and just use "D" after that. Also, perhaps avoid using 2D for two-dimensional, just spell it out to avoid confusion.
43. ...are shown for \gamma=0 and \gamma=25 deg.
44. consistency in your units... m s^-1 instead of m/s

**Section 5**

45. As summarized before, the estimation
46. tracks the wake
47. Shouldn't this be capitalized?
48. I am not sure what this means, "fits well for the application"
49. used
50. Two different cases are analyzed. In the first case, the turbine rotor is perpendicular to the wind direction (\gamma=0 deg) and these results are shown in Figure 8. In the second case, the yaw misaligment is 30 deg so that the wake is deflected.
51. are
52. to estimate
53. its
54. parameters
55. example
56. D

**Section 6**

57. repetitive, please reword, e.g. the Smith Predictor approach, which is based on an internal model control, has beenderived and used in many applications. Also combine paragraphs 1 and 2 here (lines 2-6)
58. It consists of a proportional-integral controller.
59. what is approximated with a delay? sentence has no subject?
60. main wind direction? or mean?
61. induced?
62. I think you already defined c_t up above so no need to spell out thrust coefficient?
63. define PI up above when you first talk about it
64. precludes

65. measurement
66. what is the complementary sensitivity?

**Section 7**

67. You might want to mention that your work is unique because you look at the very near wake, which is quite challenging, while other work has mostly focused on d>=2 D?
68. redirection

[revised manuscript text omitted]